# Enteral Bioactive Factor Supplementation in Preterm Infants: A Systematic Review

**DOI:** 10.3390/nu12102916

**Published:** 2020-09-24

**Authors:** Elise Mank, Eva F. G. Naninck, Jacqueline Limpens, Letty van Toledo, Johannes B. van Goudoever, Chris H. P. van den Akker

**Affiliations:** 1Department of Pediatrics-Neonatology, Emma Children’s Hospital, Amsterdam UMC, University of Amsterdam, Vrije Universiteit, Meibergdreef 9, 1105 AZ Amsterdam, The Netherlands; e.mank@amsterdamumc.nl (E.M.); e.f.g.naninck@uva.nl (E.F.G.N.); l.vantoledo@amsterdamumc.nl (L.v.T.); h.vangoudoever@amsterdamumc.nl (J.B.v.G.); 2Swammerdam Institute for Life Sciences-Center for Neuroscience, University of Amsterdam, Science Park 904, 1098 XH Amsterdam, The Netherlands; 3Medical Library, Amsterdam UMC, University of Amsterdam, Meibergdreef 9, 1105 AZ Amsterdam, The Netherlands; c.e.limpens@amsterdamumc.nl

**Keywords:** trophic factor, premature neonate, human milk, nutrition

## Abstract

Feeding preterm infants with mother’s own milk is associated with a reduction in postnatal complications and an improved neurocognitive outcome. Therefore, the bioactive factor composition of human milk has been used as a tool for the development of nutritional supplements with a potential prophylactic or therapeutic effect. The aim of this systematic review was to provide an overview on bioactive factors which have been studied as supplement to enteral nutrition in randomized controlled trials, and to provide an overview of ongoing trials. MEDLINE, EMBASE, CENTRAL, and clinical trial registers were searched. Studies on the antimicrobial protein lactoferrin were excluded as these were summarized very recently in three separate systematic reviews. Studies on vitamins D, K and iron were also excluded as they are already incorporated in most international guidelines. We identified 17 different bioactive factors, which were investigated in 26 studies. Despite the encouraging potential effects of several bioactive factors, more high-quality studies with a sufficient number of preterm infants are required before a certain factor may be implemented into clinical practice. Three large trials (*n* > 500) that investigate the effects of either enteral insulin or vitamin A are currently ongoing and could provide more definite answers on these specific supplements.

## 1. Introduction

Preterm infants are vulnerable for postnatal complications, such as necrotizing enterocolitis (NEC), sepsis, bronchopulmonary dysplasia (BPD), retinopathy of prematurity (ROP), and neurodevelopmental impairment [1]. Feeding preterm infants with mother’s own milk is associated with a reduction in postnatal complications and an improved neurocognitive outcome [2,3]. Therefore, leading health authorities such as the World Health Organization and the European Society for Pediatric Gastroenterology, Hepatology and Nutrition highly recommend mother’s own milk as first choice for preterm infants [4,5].

The protective effect of own mother’s milk is merely attributed to its numerous non-nutritive bioactive factors with various biological functions. Some bioactive factors (e.g., hormones and growth factors) mediate improved nutrient absorption by interacting with intestinal cells and modulating their growth and differentiation, resulting in an accelerated gastrointestinal tract maturation [6]. Other bioactive factors in human milk (e.g., immunoglobulins, cytokines, nucleotides, interleukins, lactoferrin, prebiotics, probiotics, and zinc) have an immune modulating effect, thereby protecting against pathogens [7]. A third group of factors form functional building blocks in cellular membranes (e.g., sphingolipids, cholesterol and fatty acids) and are for example required for neurodevelopment [8]. Moreover, a synergistic action seems to exist between several bioactive factors [9]. In general, concentrations of bioactive factors are highest in colostrum and decrease over the course of lactation [6]. In contrast to mother’s own milk, most bioactive factors are currently absent in formula feeding, and some factors are reduced in donor human milk due to multiple freeze-thawing cycles, container changes, and heat required pasteurization processes [10].

On the basis of its health benefits, providing these bioactive factors as an additive to formula or human milk might accelerate postnatal maturation and modulate the immune response beneficially. Ultimately, this could ameliorate short- and long-term outcomes of preterm infants.

To date, the majority of the clinical trials regarding enteral supplementation of bioactive factors has focused on the efficacy and safety of the antimicrobial protein lactoferrin. These data have been summarized in very recently published systematic reviews [11,12,13]. The aim of this systematic review was to provide an overview and guidance of other well-known bioactive factors, which have been studied as an enteral supplement, and to provide an overview of ongoing trials.

## 2. Materials and Methods

### 2.1. Protocol Registration and Guidelines

This systematic review was registered in PROSPERO (International Prospective Registry of Systematic Reviews), registration number: CRD42020163062. We conducted and reported the review according to the Preferred Reporting Items for Systematic Reviews and Meta-analyses (PRISMA) guidelines.

### 2.2. Eligibility Criteria

Studies were included if they met all of the following inclusion criteria: a randomized controlled trial (RCT), including preterm infants (gestational age (GA) < 37 weeks) who were enterally fed with mother’s own milk, and/or donor human milk, and/or formula. The intervention should include enteral administration of bioactive factor(s), initiated within the first month of life, and should be terminated before or at discharge from hospital. A bioactive factor or compound was defined as a molecule that is also naturally present in human milk and can be separately administered in a similar form (e.g., after chemical synthesis or recombinant DNA techniques).

Studies were excluded when they concerned parenteral/intramuscular/subcutaneous administration of the investigated bioactive factor followed by enteral administration or when the investigated bioactive factor was a macronutrient (including specific amino acids or fatty acids) or electrolyte. We excluded RCTs investigating the effect of lactoferrin, as systematic reviews of these factors have been published very recently [11,12,13]. In addition, we excluded enteral bioactive factor supplements that are administered as part of a human milk fortifier and those that are already incorporated in many international guidelines for preterm infants (i.e., vitamin D, vitamin K, and iron). Although human milk may contain a microbiome in itself [14], we excluded studies on probiotics as well as these have been summarized and discussed in detail elsewhere [15,16]. We applied relatively arbitrarily a date restriction for studies published before 2000, in order to include only studies that reflect current neonatal clinical care at best. No language restrictions were applied.

### 2.3. Outcomes

The main outcomes were number of postnatal days required to achieve full enteral feeding, NEC (according to Bell’s criteria), blood-culture proven late-onset sepsis, and mortality. Additional outcomes were clinical or suspected late-onset sepsis, meningitis, BPD (defined as oxygen requirement at 36 weeks postmenstrual age (PMA)), ROP, intraventricular hemorrhage (IVH), urinary tract infection, duration of hospital stay, intestinal maturation (intestinal permeability and lactase activity measured by sugar absorption tests), anthropometry (until 2 years of corrected age), and neurodevelopmental outcomes (until 2 years of corrected age). The safety outcomes were antibody development against the enterally supplemented bioactive factor and other clinical adverse events.

### 2.4. Search Strategy

An information specialist (J.L.) performed a systematic literature search in OVID Medline, OVID Embase and the Cochrane Controlled Register of Trials (CENTRAL) using controlled terms, i.e., MESH-terms and text-words for (1) premature/low birth weight infants and (2) enteral supplementation/bottle feeding or specific enteral bioactive factors (Appendix A). The search was combined with a methodological filter to identify RCTs. The databases were searched in April 2020.

Conference abstracts were included. We also searched the clinical trial registers ClinicalTrials.gov and WHO_ICTRP. Identified records were imported into Endnote, and duplicate records were removed. Reference lists and citing articles of identified relevant papers were crosschecked for additional relevant studies using Web of Science, and the search strategy was adapted in case of additional relevant records. 

### 2.5. Study Selection

Two reviewers (E.M. and E.N.) independently assessed all potentially eligible studies based on title and abstract using Rayyan as a webtool. The full-texts of the potentially eligible papers were obtained and assessed for eligibility by the same two reviewers. Disagreement between reviewers was resolved through discussion or by consultation with a third author (C.v.d.A.).

### 2.6. Data Extraction

A standardized form was used to extract data from the included studies for evidence synthesis. The extracted information included: authors, year of publication, country, methodology, study design, definition of clinical outcomes, measurement tools to quantify outcomes, sample size, type of bioactive factor, dosage, intervention duration, day of study initiation, in- and exclusion criteria, participants’ characteristics, follow-up duration, number of participants included in the analysis, number of withdrawals, number of exclusions, number of lost to follow-up, and patient outcomes. Authors were contacted twice at maximum when relevant data were not (or not clearly) reported. Growth velocity during the study period was calculated if data was available on weights at baseline and at the study end together with the precise duration of the study period. Growth in g/kg per day was calculated according to Patel et al. [17]: (1000 × ln(W_n_/W_1_))/(D_n_ − D_1_). W_1_ is the average weight at the beginning of the study period; W_n_ is the average weight at the end of the study period, and D_n_ − D_1_ is the study period.

### 2.7. Risk of Bias Assessment

Risk of bias of the included studies was assessed independently by two reviewers (E.M. and E.N.) using the revised Cochrane Collaboration’s risk of bias tool (version 2.0, August 2019) [18]. The tool included the following items: randomization process, deviations from intended interventions, missing outcome data, measurement of the outcome, and selection of the reported result. According to the risk of bias assessment, reviewers independently categorized the study’s risk of bias as low, some concerns, or high. Disagreement between reviewers was resolved through discussion or by consultation with a third author (C.v.d.A.).

### 2.8. Statistical Analysis

Variables are expressed as mean ± standard deviation (SD), median (interquartile range [IQR]), or as frequency. Categorical data were analyzed by calculating the odds ratios. Meta-analysis was performed by using Review Manager (version 5.4, The Cochrane Collaboration, Copenhagen, Denmark) if at least three studies investigated a similar product and further study design was similar enough to do so. We used a more conservative random-effects model.

## 3. Results

### 3.1. Study Selection

The study selection process is described in Figure 1. The search strategy identified 1571 unique records, of which 1450 were excluded based on title and/or abstract. A total of 97 full texts were screened, of which 24 met our inclusion criteria and reported usable extractable outcome data. Two additional studies were identified via cross-references. The main characteristics and study details of the 26 included studies are shown in Table 1.

### 3.2. Risk of Bias Assessment

Five trials (19%) were judged as with an overall high risk of bias, based on various subdomains (Figure 2) [21,25,26,31,44]. Risk of bias assessment revealed some concerns for 13 trials (50%), as they were not preregistered in trial registries, and no predefined statistical analysis plan and/or study protocol was published for these studies, which precluded selective data reporting judgement. However, one should consider that before the year ~2005–2010, awareness regarding preregistering trials with predefined statistical plans was not ubiquitous. Three of these 13 studies also did not provide complete information about the randomization process either [22,24,41]. Eight trials (31%) were of low risk of bias, all published between 2013 and 2020.

### 3.3. Hormones and Growth Factors

Enteral hormones and growth factors interact with intestinal cells in the neonatal intestine and modulate their growth and differentiation, resulting in an accelerated gastrointestinal development [6].

#### 3.3.1. Prophylactic Supplement

We identified six studies investigating the effect of a recombinant hormone or growth factor as an enteral supplement in order to prevent prematurity-related complications [19,20,21,22,23,24]. Four of these studies tried to establish an effect of providing recombinant human erythropoietin (rhEPO) and/or recombinant human granulocyte colony-stimulating factor (rhG-CSF). In the first study, by El-Ganzoury et al. [19], 90 preterm infants with a GA of ≤33 weeks were randomly assigned between four groups: 20 received 4.5 µg/kg/day rhG-CSF, 20 received rhEPO (88 IU/kg/day), 20 received both interventions simultaneously, and 30 received only placebo, all until an enteral intake of 100 mL/kg/day was reached or for a maximum of 7 days. Treatment with rhEPO, rhG-CSF, or both resulted in a significantly shorter time to full enteral feeding and a shorter duration of hospital stay (Table 2). The incidence of NEC stage 3 amounted 10% in the placebo group, whereas none of the infants in the two intervention groups suffered from NEC stage 3, though this was not statistically significant. In another study, by Omar et al. [20], partly similar in design, only 88 IU/kg/day rhEPO or placebo was administered to preterm infants (GA ≤ 32 weeks, *n* = 120). In this study, however, no beneficial effect on time to achieve full enteral feeding or NEC stage ≥2 was observed when compared to the placebo group (Table 2).

In the study by Hosseini et al. [21], preterm infants were randomly assigned to receive either 5 mL/kg/day of synthetic amniotic fluid containing 225 ng/mL rhG-CSF (*n* = 50), 5 mL/kg/day of synthetic amniotic fluid containing both 225 ng/mL rhG-CSF and 4400 µU/mL rhEPO (*n* = 50), or standard care (*n* = 50). Other components of the synthetic amniotic fluid were electrolytes and albumin. The incidences of NEC stage ≥2 (Table 2) or NEC requiring surgery(rhG-CSF group: 1 (2%), rhG-CSF + rhEPO group and placebo group: 0 (0%), *p* = 0.859) were not different between the groups. Mortality rate was significantly lower in both intervention groups compared to the placebo group. The study was judged as high risk of bias as the randomization sequences seemed not concealed and NEC assessment might have been influenced by knowledge of the intervention as the study was not blinded.

The highest rhEPO dosage (1000 IU/kg/day) was administered in the study by Juul et al. [22]. The NEC rates and growth velocities were not significantly different between the intervention (*n* = 15) and control (*n* = 17) groups (Table 2).

Corpeleijn et al. [23] randomized 60 preterm infants (BW 750–1250 g) to bovine insulin-like growth factor 1 (IGF-1) enriched preterm formula or standard preterm formula for 4 weeks. Time to achieve full enteral feeding, growth velocity, and duration of stay in the NICU were not significantly different between the groups (Table 2). In addition, no effects were observed on morbidity and mortality rates. Intestinal permeability was, however, significantly improved in the IGF-1 supplemented group on postnatal day 14 compared to the control group, while no difference was observed on days 7, 21 and 28. No significant effect of IGF-1 enriched formula on lactase activity was observed on any study day.

Shamir et al. [24] provided eight preterm infants with either enteral rh-insulin (400 µU/mL milk) or placebo (both added to formula) for 28 days. The average age at enrollment was day 5 postpartum in the intervention group and day 6 postpartum in the placebo group. A trend towards a shorter time to achieve full enteral feeding was observed (Table 2). Weight gain (total grams) during the intervention was significantly higher in the intervention group compared to the control group (*p* < 0.02); however, growth velocity (in g/kg/day) did not reach a statistically significant effect. Two trials with larger sample sizes (first study: *n* = 33, second study: *n* = 530), investigating the effects of enteral rh-insulin, are currently ongoing (Table 3).

#### 3.3.2. Therapeutic Supplement

The effects of any hormone or growth factor as therapeutic supplement in preterm infants was assessed in two studies [25,26]. The first study, by El-Kabbany et al. [25], was conducted in 40 preterm infants (GA < 36 weeks) with early or late-onset sepsis (based on clinical signs and laboratory indicators including Rodwell’s hematological score ≥3). Participants received a single dose of melatonin (20 mg) or placebo. A significantly lower mortality rate was observed in melatonin-treated preterm infants, compared to the placebo group (Table 2). Furthermore, length of hospital stay was reported as a median of only 3 days in the melatonin group. This seems unrealistic, and in the absence of a reply by the authors, it remains speculative whether weeks instead of days were denoted, or whether it was referred as days on intensive care. Besides, the study was judged as high risk of bias, as the randomization sequences was not concealed, and the study was not blinded.

The second study, by Canpolat et al. [26], was conducted in 18 preterm infants with NEC stage 1. None of the infants in the intervention group (20 µg/kg/day rhG-CSF for 5 days) showed disease progression, while 5 (50%) infants in the placebo group progressed to NEC stage 2 or 3 (*p* < 0.05). In addition, duration of hospitalization was significantly shorter in the rhG-CSF treated group (Table 2). However, it must be noted that this study was rated as having a high risk of bias, as the study seemed not blinded, and NEC progression assessment might have been influenced by knowledge of the intervention.

### 3.4. Vitamins

Vitamin A regulates growth and differentiation of epithelial cells, especially in the respiratory tract [47]. In addition, vitamin A is an important radical-scavenging antioxidant and has a regulatory role in the functioning of immune cells [47]. Several trials investigated the effects of extra supplementation in premature infants. Sun et al. randomly assigned 262 preterm infants (GA < 28 weeks) to receive either 1500 IU/day vitamin A (retinol) or placebo enterally for 28 days or until discharge, whichever came first [27]. A significant reduction in incidence of BPD and type I (i.e., severe) ROP, as well as a reduced hospital stay duration, was observed in vitamin A treated preterm infants compared to the placebo group (Table 2). In addition, the composite outcome type I ROP or mortality occurred significantly less frequent in the intervention group (*p* = 0.03).

The effects of higher vitamin A dosages were investigated in two other studies. The first study, published in 2001 by Wardle et al. [28], did not demonstrate an effect of 5000 IU/kg/day vitamin A for 28 days on morbidity or mortality rates in 154 preterm infants with a BW < 1000 g. The second study, published in 2019 by Basu et al. [29], demonstrated a significantly lower incidence of culture-proven sepsis and a trend towards reduction of BPD (*p* = 0.05) in preterm infants (BW < 1500 g) treated with 10,000 IU vitamin A on alternate days, compared to the placebo group (*n* = 196 in total, Table 2). Meta-analysis of all studies revealed no effect of enteral vitamin A supplementation on BPD and mortality (Figure 3). No meta-analysis of other clinical outcomes was performed as the definitions differed too much between the studies.

Vitamin E is an important radical-scavenging antioxidant and has a regulatory role in the functioning of immune cells [48,49]. The effect of enteral vitamin E supplementation in preterm infants was assessed in three studies, all with a low sample size [30,31,32]. In the first study, by Bell et al. [30], a single dose (50 IU/kg) vitamin E (dl-α-tocopheryl acetate) within 4 h postnatally did not have an effect on culture-proven sepsis, NEC, IVH stage ≥3, or mortality rates in preterm infants (*n* = 93, Table 2). In the second study, by Barekatain et al. [31], 80 preterm infants (GA ≤ 30 weeks) were randomly assigned to receive 10 IU/day vitamin E (form of vitamin E unknown) or placebo for the first 3 postnatal days. The incidences of culture-proven sepsis, clinical sepsis, NEC, IVH grade ≥3, and mortality all tended to be lower in the intervention group, but none of the outcomes reached statistical significance (Table 2). In the third study, by Pathak et al. [32], 50 IU/day of vitamin E (α-tocopherol) for 8 weeks did not have an effect on culture-proven sepsis (both groups 33%), NEC, or duration of hospital stay in a small group of preterm infants (*n* = 30 in total, Table 2).

### 3.5. Carotenoids

Carotenoids are physiological macular pigments, protecting the retina from oxidative stress and related diseases [50]. The effects of carotenoid (i.e., lutein and zeaxanthin) supplements were investigated in four studies [33,34,35,36]. Two studies administered placebo or a lower dose of 0.14 mg/day lutein combined with 0.006 mg/day or 0.0006 mg/day zeaxanthin from the first week of life until 36 weeks PMA or discharge (*n* = 114 and *n* = 229) [33,34], and two studies (*n* = 77 and *n* = 63) administered placebo or a dose of 0.5 mg/kg/day lutein combined with 0.02 mg/kg/day zeaxanthin starting from the 7th day postpartum until 40 weeks PMA or discharge [35,36]. Lower incidences of severe ROP were found in three of the four studies, although rates were not statistically different [33,34,36]. One study did not consider ROP as outcome [35]. In addition, three of the four studies observed a lower—but not significantly—incidence of BPD [34,35,36]. Meta-analysis of all studies also revealed no significantly different effects of any carotenoids on either all-stage ROP, severe ROP, culture-proven sepsis, BPD, or mortality rates (Figure 4).

### 3.6. Enzymes

The effect of recombinant human bile salt-stimulated lipase (rhBSSL) was investigated by Casper et al. [37]. BSSL is a lipolytic enzyme facilitating digestion and absorption of fat in human milk and formula. A total of 415 preterm infants (GA < 32 weeks) were randomized to receive either rhBSSL or placebo added to formula or pasteurized human milk (15 mg/100 mL milk) for 4 weeks. No effects were observed on growth velocity or Bayley neurodevelopmental scores at 12 months of corrected age. Subgroup analysis, however, revealed a significantly higher growth velocity in small-for-gestational age (SGA) infants in the rhBSSL group compared to the placebo group (rhBSSL group (*n* = 32): 17.1 g/kg/day, placebo group (*n* = 30): 15.2 g/kg/day, estimated difference 1.95 (95% CI 0.38-3.52 g/kg/day). It was noted that 16.5% of infants in the intervention group developed anti-drug antibodies at some point during the study. In these infants, no hypersensitivity reactions or NEC events were observed.

### 3.7. Trace Elements

Selenium is incorporated in selenoproteins (e.g., glutathione peroxidase), which have an important anti-oxidant and antimicrobial function [51]. Therefore, the effect of enteral selenium supplements on sepsis was investigated by Aggarwal et al. [38]. They randomly assigned 114 preterm infants (GA < 32 weeks) to receive either 10 µg/day selenium or placebo until day 28 of life. Significant reductions of both culture-proven and suspected sepsis rates were observed in selenium-treated preterm infants compared to the placebo group (Table 2). In addition, duration of NICU stay was significantly shorter. No effect on ROP stage ≥2 was observed.

Zinc is a trace element with catalytic (e.g., enzyme activity), structural (e.g., cell membrane), and regulatory functions (e.g., protein synthesis and gene expression) [52,53]. The effect of supplemental zinc in preterm infants (GA 24–32 weeks or BW 401–1500 g) was investigated by Terrin et al. [39]. Participants (*n* = 193) received daily either 9 mg of zinc or placebo from the 7th day of life until 42 weeks PMA or until discharge, whichever came first. A significantly lower incidence of NEC stage 3 and mortality was found in zinc-treated preterm infants. No effects were observed on other morbidity outcomes (Table 2). Body weight at discharge was higher in the zinc group compared to the placebo group (zinc group: 2208 ± 501 g, placebo group: 1889 ± 639 g, *p* = 0.001), but daily weight gain was not significantly different. Four studies (790 infants in total) investigating the effects of zinc are currently ongoing (Table 3).

### 3.8. Compounds of Plasma Membrane Lipids

Two studies investigated the effects of adding compounds of plasma membrane lipids to enteral nutrition for preterm infants [40,41]. In the first study [40], 30 preterm infants (mean GA ~29 weeks) whose mothers did not provide human milk, were randomized to either low-dose (LD) (<0.03 g/L, standard preterm formula), medium-dose (MD) (0.15 g/L, similar to human milk), or high-dose (HD) (0.30 g/L) cholesterol fortified preterm formula until 40 weeks PMA. Average body growth and head growth velocity were highest in the MD and HD cholesterol groups, although they did not reach statistical significance (Table 2). At 40 weeks PMA, average body weights and lengths were lower in the LD compared to the MD and HD groups; however, the difference was only significant between the LD and the MD groups (LD: weight 2739 ± 372 g and length 46.0 ± 2.0 cm; MD: weight 3161 ± 347 g and length 48.2 ± 1.4 cm). Average head circumferences at 40 weeks PMA were not significantly different (LD: 34.0 ± 1.2 cm, MD 35.1 ± 1.0 cm, HD 34.7 ± 1.2 cm). No other clinical endpoints were reported in this study.

In the second study, by Tanaka et al. [41], the effects of increasing the sphingomyelin (SM) content in formula on neurodevelopmental outcome (Bayley-II score) at 6, 12, and 18 months of corrected age were investigated. Twelve preterm infants (BW < 1500 g) received SM-fortified formula (SM 20% of all phospholipids in milk), and 12 infants received the standard formula (SM 13% of all phospholipids). No effects were observed on the mental or psychomotor developmental indices. The orientation and emotional scores on the Behavior Rating Scale (BRS) at 6, 12, and 18 months were significantly higher in the enriched SM group than in the standard group (*p* < 0.05). At 12 and 18 months of corrected age, the motor quality on the BRS was significantly higher in the enriched SM group (*p* < 0.05). Body weight, length, and height at 18 months of corrected age were similar between the groups.

### 3.9. Creatine

Creatine is primarily stored in the skeletal muscle as free creatine or as its phosphorylated form and functions as major energy source [54]. The effects of supplementing creatine monohydrate were assessed by Bohnhorst et al. [42]. Late-preterm infants (GA 32–36 weeks) with symptoms of apnea of prematurity requiring treatment with caffeine (≥1 bradycardia and/or desaturation per hour or >1 apnea requiring bagging, measured over 6 consecutive hours) were randomly assigned to receive either 200 mg/kg/day creatine monohydrate or placebo for 2 weeks. No effect on growth was observed (Table 2). The authors did not report other outcomes of interest for our systematic review.

### 3.10. Immunoglobulins

Enteral immunoglobulines may potentially have an immunoprotective effect through their pathogen neutralizing capacity in the gastrointestinal tract [7]. The effect of enteral supplementation of immunoglobulin G (IgG) was assessed by Lawrence et al. [43]. A total of 1529 preterm infants (BW ≤ 1500 g) were randomly assigned to 1200 mg IgG/kg/day or placebo for 28 days. No effects on incidences of NEC (IgG group: 7%, placebo group: 6%) or mortality (both groups: 3%) were observed. Sepsis rates were not reported.

### 3.11. Nucleotides

Nucleotides have a role in many biological processes as they are building blocks of DNA and RNA [55]. No effect of nucleotide enriched formula from postnatal days 2 to 16 on growth velocity in 14 preterm infants (GA 30–37 weeks) were observed by Scopesi et al. [44] (Table 2). The study was judged as high risk of bias as information about the randomization process was partly missing, and the study was not blinded.

## 4. Discussion

To date, accumulating knowledge on the naturally existing bioactive factors in human milk provides important insights for the development of nutritional supplements with a potential prophylactic or therapeutic effect, especially for preterm infants [6]. Quite a number of different bioactive factors have been researched as supplement to enteral nutrition in preterm infants as described in this systematic review. Unfortunately, most were evaluated in only one or two trials each. In addition, only 11 studies included more than 100 patients in total (of which only 3 studies more than 250 infants), so that statistical power to detect differences in meaningful clinical outcomes was very low for most factors. To illustrate, in order to demonstrate a hypothetical optimistic reduction in NEC incidence from for example 10% to 5% (α of 0.5 and power of 80%), 434 infants per group would be needed.

In our review, the group of recombinant hormones and growth factors as a prophylactic or therapeutic supplement in preterm infants was studied most frequently, in 8 of the 26 studies. Nonetheless, it is difficult to draw conclusions to date, as for example enteral administration of insulin was only described in a pilot study with eight preterm infants [24]. Two larger randomized controlled trials with 563 preterm infants are currently ongoing and will provide more meaningful answers as to whether enteral insulin could exert effects on short- and long-term outcomes (Table 3).

No effect of bovine IGF-1 supplementation on time to full enteral feeding was observed [23]. The authors suggested that the lack of efficacy could also have been caused by their low administered IGF-1 dosage in the first postnatal days, as the IGF-1 was added to the formula (10 µg/100 mL) which is only tolerated in low volumes in the first days after birth. The bovine origin of IGF-1 instead of a recombinant human product could be a drawback as well. The effect of a recombinant human supplement and in higher dosages needs to be investigated in future trials. Intravenous supplementation of rhIGF-1 together with its binding protein resulted in a recent phase 2 trial in a reduction of BPD rates [56]. Currently, a phase 3 study is enrolling patients.

The effect of enteral rhEPO was investigated in four studies [19,20,21,22]. None of these studies showed an significant NEC incidence reduction, which is in line with very recently published results of intravenous supplementation of high-dose rhEPO (1000 IU/kg every 48 h for a total of six doses) in a large sample size (*n* = 941) of preterm infants (GA 24–28 weeks) [57]. The effect of enteral rhEPO on time until full enteral feeding was contradictory, which might be caused by the limited sample size of the studies and usage of different brands rhEPO [19,20]. Additional studies are required to provide more conclusive results.

Most vitamins are routinely added in certain amounts to milk for preterm infants, through the use of human milk fortifiers. In addition, vitamin D and K supplements are incorporated in most international guidelines to further counteract deficiencies in neonates. Extra vitamin A supplementation has been a focus of interest, as it was shown that intramuscular injections with vitamin A (three times a week) may result in lower BPD and all-stage ROP rates, although the overall quality of the evidence was moderate, and the administered dosages differed widely [58]. Moreover, intramuscular injections are invasive, and therefore, enteral administration would form an attractive route of administration.

To date, three studies assessed the effects of enteral administration of vitamin A. Interestingly, a significant reduction in BPD and severe ROP rates were observed only in the study investigating the lowest dosage of vitamin A [27]. In addition, the authors described even a reduction in total hospital stay of one month in infants in the intervention group. A higher dosage was administered in two other studies [28,29]. Of these, one study found a borderline significantly lower BPD incidence in the intervention group (*p* = 0.05) [29], while the other study did not observe any effect, possibly also due to the fact that many infants (21–26%) in both groups received postnatal steroids [28]. Thus, drawing definite conclusions is hampered by the relatively small sample sizes despite meta-analysis in Figure 3 showing positive overall trends in reducing BPD or mortality. Two studies with a large sample size (~900 infants in each study) and two studies with moderate sample size (~200 infants in each study) are currently ongoing and will hopefully provide more meaningful answers (Table 3).

Administration of vitamin E for 1 day, 3 days, or 8 weeks did not have an effect of morbidity or mortality outcomes [30,31,32]. The serum α-tocopherol levels might have been too low to reveal any effect. It has been shown in a meta-analysis summarizing studies from the 1980s that serum concentrations greater than 3.5 mg/dL were required to prevent IVH and ROP [59]. However, caution is warranted as these higher serum levels were simultaneously accompanied with a higher sepsis incidence [59]. In the study by Bell et al. [30], one-third of the infants had a serum α-tocopherol concentration between 0.5–3.5 mg/dL, and one-third <0.5 mg/dL measured 24 h after a single dose dl-α-tocopheryl acetate. In the study by Pathak et al. [32], the mean serum α-tocopherol concentration in infants receiving α-tocopherol was 2.1 mg/dL in the third week and 2.9 mg/dL in week 8 of the intervention.

In four studies, enteral supplementation of carotenoids seemed to result in a lower incidence of oxidative-stress induced complications of prematurity, although none were statistically different [33,34,35,36]. An explanation for the lack of efficacy of carotenoid supplementation could have been the demonstrated lack of significant differences in plasma concentrations between the intervention and placebo groups [35,36], despite it being shown elsewhere that lutein is well absorbed in the preterm intestine [60]. Similar plasma concentrations might have been caused by the administration of mother’s milk in both study groups, as mother’s milk is a source of carotenoids that seems to have a higher bioavailability compared to carotenoids in a pharmaceutical formulation [61]. In addition, the size of the study population (*n* = 483 in total) might have been too small to observe any significant effect.

The addition of rhBSSL to formula or donor human milk did not result in a higher growth velocity in a phase 3 study [37]. This result is in contrast with the phase 2 study, which had a cross-over design with a washout period of 2 days (*n* = 65 in total) [62]. The contradictory results might be caused by the different designs of the studies. In addition, the growth rates in the placebo group of the phase 3 study (rhBSSL group: 16.8 g/kg/day, placebo group: 16.6 g/kg/day) was higher than during the phase 2 study (rhBSSL group: 16.9 g/kg/day, placebo group: 13.9 g/kg/day), which might have concealed the effect of rhBSSL. The authors suggested that the higher growth rate in the phase 3 study was possibly caused by improved nutritional management in the years between the phase 2 and 3 studies. Subgroup analysis of the phase 3 study, however, still revealed a significantly higher growth velocity in SGA infants.

Selenium supplementation had a significant effect on culture-proven and suspected sepsis, but not on the incidence of ROP, despite the antioxidant-enhancing properties of selenium as part of the glutathione-peroxidase enzyme [38]. A higher dosage regimen might be required to obtain an adequate increase in glutathione-peroxidase activity for the prevention of ROP, as it has been shown that enzyme activity increased only slightly despite increases in serum selenium concentrations during supplementation [63]. The results are in line with a meta-analysis, which revealed no effect of intravenous selenium supplementation on ROP, while the sepsis rate was significantly reduced as well [64].

By far, the largest study in this systematic review was on enteral supplementation of IgG [43]. In 1529 infants, no effect was shown on the NEC incidence. Unfortunately, sepsis rates were not assessed. These results are in line with a recently published meta-analysis summarizing the effects of intravenously administered IgG [65]. It remains to be speculated whether IgA supplementation, for example, would result in a lower NEC or sepsis incidence, as human milk contains far more (secretory) IgA than IgG [7].

Overall, none of the authors of the included studies described adverse clinical events related to their administered bioactive factor. One exception could be the study by Casper et al. [37], who noted a higher NEC incidence in rhBSSL treated infants (3.4%) compared to the placebo group (0.5%). As both incidences are regarded as low, the clinical relevance remains unsure. Furthermore, 16.5% of infants in the intervention group developed anti-drug antibodies at some point in that particular study. However, none of these infants showed hypersensitivity reactions or suffered from NEC.

The effects of enteral lactoferrin were outside our scope as these were described very recently in systematic reviews elsewhere [11,12,13]. To summarize, enteral bovine or recombinant human lactoferrin resulted in a possibly slightly lower incidence of late-onset sepsis (studied in 5425 infants in total), but no effects were observed on NEC, mortality, or neurodevelopment [11]. Besides, there was high heterogeneity between studies, and no effects were shown in the two largest studies (>75% of evidence base). Eight more RCTs in 2580 infants on lactoferrin supplementation are currently underway (Appendix A).

This systematic review summarizes 26 studies on several bioactive factors. Human milk, however, contains many more bioactive factors that have not been investigated to date, such as adipokines or anti-inflammatory cytokines [6]. Moreover, the provision of human milk oligosaccharides to preterm infants has not been studied yet in RCTs. A recent systematic review on mimicking prebiotic fibers (mainly galacto- and fructo-oligosaccharides) showed a reduction in several neonatal morbidities, except for NEC [66].

There is thus much room for additional biological factors to be investigated. Besides, nearly all studies described here investigated the effects of a single bioactive factor, while a synergistic action between several bioactive factors may be required to sort out its effects. One interesting strategy recently described is the use of bovine colostrum as a fortifier for mother’s milk or as a (partial) replacement for formula as it contains many active and potentially protective bioactive factors [67,68,69]. Four studies investigating the effects of bovine colostrum are currently ongoing (Appendix A in Supplemental Materials).

## 5. Conclusions

This systematic review for which we employed an extensive search strategy, identified 17 bioactive factors, next to lactoferrin, which have been studied as an enteral supplement. Unfortunately, most are only studied in highly underpowered trials, often also not free from several biases. Besides, the number of studies investigating a similar bioactive factor was limited, and if so, there was a high heterogeneity between study designs and outcomes, hampering meta-analysis for most bioactive factors. Despite the encouraging potential effects of several bioactive factors as described here, more high-quality studies with a sufficient number of preterm infants are required before a certain factor is recommended to be implemented into clinical practice. Three large trials (*n* > 500 each study), in which the effects of either enteral insulin or vitamin A are investigated, and four trials (*n* = 790 in total) on enteral zinc are currently ongoing and could provide more meaningful answers on these supplements.

## Figures and Tables

**Figure 1 nutrients-12-02916-f001:**
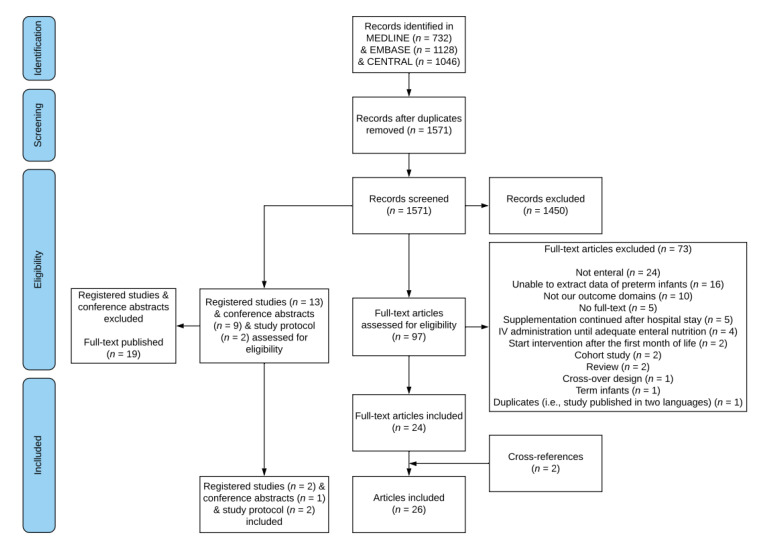
Flowchart of the study selection process.

**Figure 2 nutrients-12-02916-f002:**
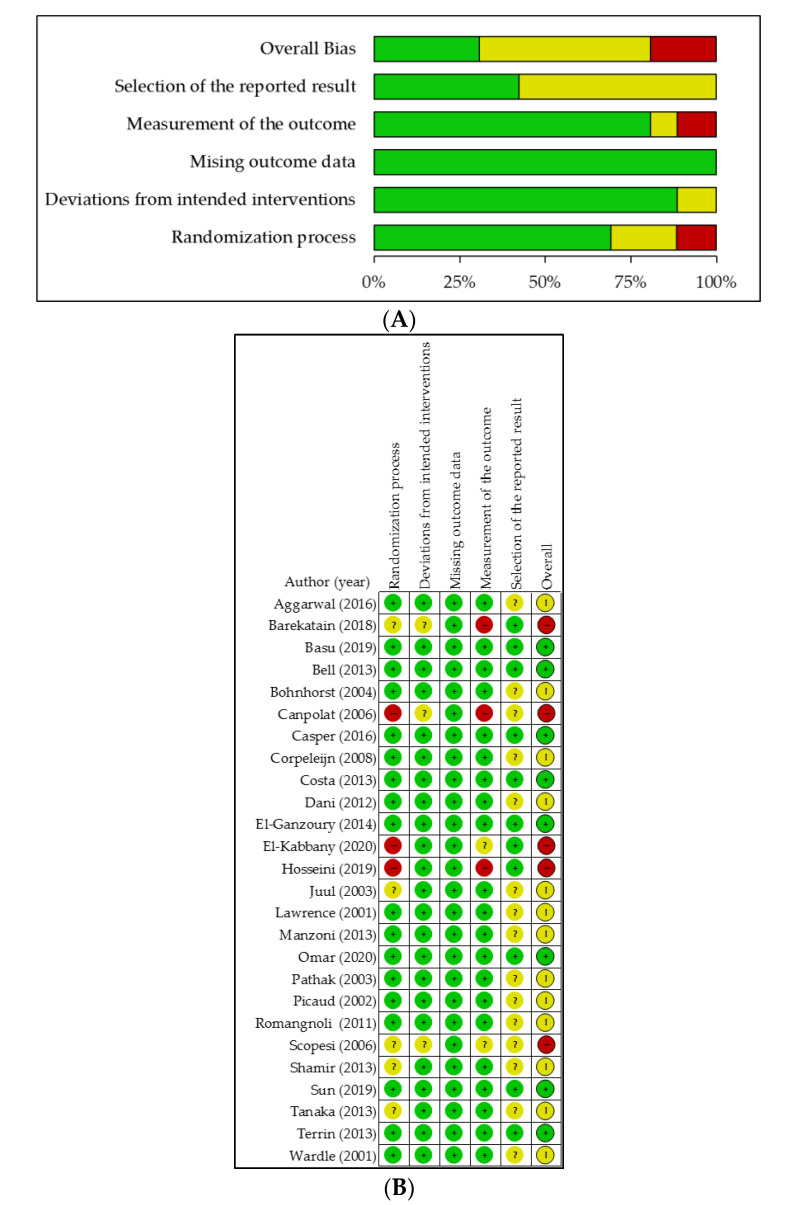
Review author’s risk of bias judgement, presented as percentage across all included studies at each level of risk of bias (**A**) and for each included study (**B**). Green: low risk, yellow: some concerns, red: high risk.

**Figure 3 nutrients-12-02916-f003:**
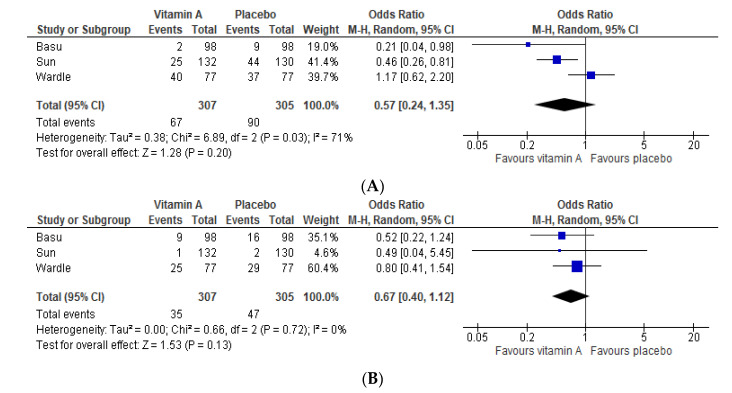
Effect of any enteral vitamin A supplement on BPD (**A**) and mortality (**B**). BPD: bronchopulmonary dysplasia.

**Figure 4 nutrients-12-02916-f004:**
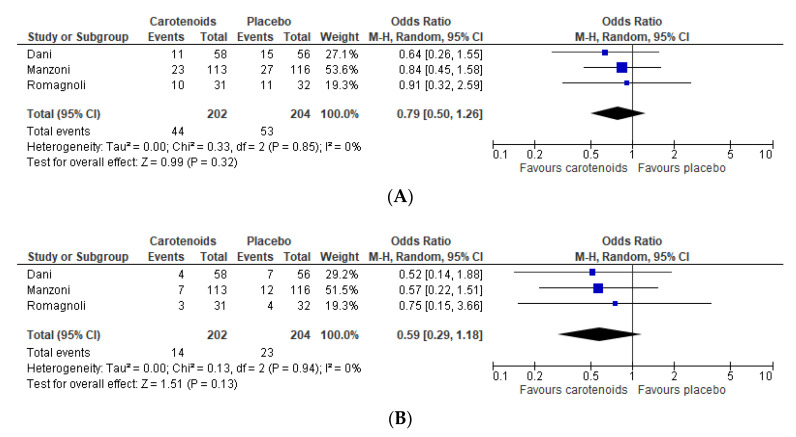
Effect of any enteral carotenoids supplement on all-stage ROP (**A**), severe ROP (**B**), culture-proven late-onset sepsis (**C**), BPD (**D**), and mortality (**E**) in preterm infants. ROP: retinopathy of prematurity.

**Table 1 nutrients-12-02916-t001:** Overview of the included studies.

First Author, Year, Country [Reference]	Main Inclusion Criteria	Number of Randomized Infants	Bioactive Factor	Initiation and Duration of the Study Intervention	Gestational Age, Weeks *	Birth Weight, g *	Primary Outcome(s)	Secondary Outcome(s)	Other Reported Outcome(s)
**Hormones and growth factors**									
El-Ganzoury, 2014, Egypt [19]	GA ≤ 33 weeks	- 20 - 20 - 20 - 30	- RhG-CSF (4.5 µg/kg/day) - RhEPO (88 IU/kg/day) - RhG-CSF (4.5 µg/kg/day) and EPO (88 IU/kg/day) - Placebo	Start: on the day that enteral feeding was initiated, duration: until an enteral intake of 100 ml/kg/day, or for a maximum of 7 days, whichever came first.	- 30.8 ± 1.8 - 30.2 ± 1.8 - 30.4 ± 1.9 - 30.5 ± 1.5	- 1260 ± 270 - 1310 ± 310 - 1190 ± 280 - 1360 ± 290	Time to full enteral feeding, NEC	-	Mortality, duration of hospital stay
Omar, 2020, Egypt [20]	GA ≤ 32 weeks	- 60 - 60	- RhEPO (88 IU/kg/day) - Placebo	Start: on the day that enteral nutrition was initiated, duration: until an enteral intake of 150 ml/kg/day, or for a maximum of 10 days, whichever came first.	- 32.0 (31.0–32.0) - 32.0 (30.5–32.0)	- -	-	Time to full enteral feeding, growth velocity, NEC	Mortality
Hosseini, 2019, Iran [21]	GA ≤ 28 weeks and BW < 1250 g	- 50 - 50 - 50	- Synthetic amniotic fluid containing rhG-CSF (225 ng/mL) - Synthetic amniotic fluid (5 mL/kg/day) containing rhG-CSF (225 ng/mL) and rhEPO (4400 µU/mL) - Standard care	Start: day 3 postpartum, duration: 21 days	- 27.7 ± 1.7 - 28.7 ± 2.6 - 27.7 ± 1.5	- 948 ± 179 - 1065 ± 189 - 998 ± 173	NEC	Sepsis, ROP, IVH, mortality	-
Juul, 2003, USA [22]	BW 700–1500 g	- 15 - 17	- RhEPO (1000 IU/kg/day) - Placebo	Start: at least 30 mL/kg/day enteral nutrition, duration: 2 weeks	- 27.8 ± 1.8 - 28.8 ± 2.1	- 1070 ± 255 - 1145 ± 259	-	-	Growth velocity, NEC
Corpeleijn, 2008, The Netherlands [23]	BW 750–1250 g	- 50 - 49	- Formula with bovine IGF-1 (10 µg/100 mL) - Standard formula	Start: day 3-7 postpartum, duration: 4 weeks	- 29 (26-33) ^#^ - 28 (25-33)	- 975 (750-1250) ^#^ - 920 (750-1250)	Time to full enteral feeding, growth velocity	Intestinal permeability, lactase activity	NEC, IVH, mortality, duration of stay at the NICU
Shamir, 2013, Israel [24]	GA 26–33 weeks and BW > 750 g	- 4 - 4	- Rh-insulin (400 µU/mL milk) - Placebo	Duration: 28 days	-	- 1285 (1058-1570) - 1205 (836-1619)	Time to full enteral feeding, growth velocity	-	-
El-Kabbany, 2020, Egypt [25]	GA < 36 weeks, sepsis	- 20 - 20	- Melatonin (20 mg) - Standard care	Single dose	- 32.4 ± 2.6 - 32.9 ± 2.1	-	-	Mortality, duration of hospital stay	-
Canpolat, 2006, Turkey [26]	GA ≤ 34 weeks, stage I NEC	- 8 - 10	- RhG-CSF (20 µg/kg/day) - Placebo	Start: first day of NEC diagnosis, duration: 5 days	- 30.8 ± 1.6 - 30.9 ± 1.7	- 1269 ± 394 - 1206 ± 280	NEC severity and progression	-	Duration of hospital stay
**Vitamins**									
Sun, 2019, China [27]	GA < 28 weeks, <96 h of age	- 132 - 130	- Vitamin A (1500 IU/day) - Placebo	Start: on the day that enteral feeding was initiated, duration: 28 days or until discharge, whichever came first	- 26.8 ± 1.9 - 27.1 ± 2.0	- 982 ± 234 - 984 ± 217	Composite of mortality or type I ROP, BPD	Sepsis, NEC, ROP, IVH, mortality, duration of hospital stay	-
Wardle, 2001, United Kingdom [28]	BW < 1000 g	- 77 - 77	- Vitamin A (5000 IU/kg/day) - Placebo	Start: day 1 postpartum, duration: up to and including day 28 postpartum	- 26 (25–27) - 26 (25–27)	- 806 (710–890) - 782 (662–880)	-	NEC, BPD, ROP, IVH, mortality	-
Basu, 2019, India [29]	BW < 1500 g, requiring invasive or non-invasive respiratory support at 24h of life	- 98 - 98	- Vitamin A (10,000 IU on alternate days) - Placebo	Start: 24h postpartum, duration: 28 days	- 30.9 ± 2.9 - 30.7 ± 2.7	- 1185 ± 194 - 1163 ± 181	-	Sepsis, NEC, BPD, ROP, IVH, mortality	-
Bell, 2013, USA [30]	GA < 27 weeks and BW < 1000 g	- 62 - 31	- Vitamin E (dl-α-tocopheryl acetate) (50 IU/kg) - Placebo	Single dose within <4 h	- 25 (24–26) - 25 (24–26)	- 700 (610–840) - 680 (600–850)	-	-	Sepsis, NEC, IVH, mortality
Barekatain, 2018, Iran [31]	GA ≤ 30 weeks	- 40 - 40	- Vitamin E (form not reported) (10 IU/day) - Placebo	Start: day of birth, duration: 3 days	- 28.4 ± 1.5 - 28.6 ± 1.4	- 1174 ± 162 - 1192 ± 176	IVH	Sepsis, NEC	Mortality
Pathak, 2003, USA [32]	GA ≤ 32 weeks and BW ≤ 1250 g	- 15 - 15	- Vitamin E (α-tocopherol) (50 IU/day) - Placebo	Duration: 8 weeks or until discharge, whichever came first	- 27.9 ± 1.9 - 27.9 ± 1.5	- 928 (170) - 899 (159)	-	-	Sepsis, NEC, duration of hospital stay
**Carotenoids**									
Dani, 2012, Italy [33]	GA ≤ 32 weeks	- 58 - 56	- Lutein (0.14 mg/day) and zeaxanthin (0.006 mg/day) - Placebo	Start: day 1-7 postpartum, duration: until discharge	- 28.8 ± 2.4 - 28.3 ± 2.4	- 1130 ± 330 - 1130 ± 360	ROP	ROP requiring laser- or cryo-treatment, sepsis, NEC, BPD, IVH, mortality	-
Manzoni, 2013, Italy [34]	GA < 32 + 6 weeks	- 113 - 116	- Lutein (0.14 mg/day) and Zeaxanthin (0.0006 mg/day) - Placebo	Start: <48 h postpartum, duration: until 36 weeks PMA	- 30.1 ± 1.8 - 29.7 ± 2.6	- 1336 ± 417 - 1271 ± 386	Threshold ROP, NEC stage ≥2, BPD	ROP of all stages, NEC of all stages, sepsis, IVH, mortality	-
Costa, 2013, Italy [35]	GA ≤ 34 weeks	- 38 - 39	- Lutein (0.5 mg/kg/day) and zeaxanthin (0.02 mg/kg/day) - Placebo	Start: day 7 postpartum, duration: until 40 weeks PMA or until discharge, whichever came first	- 30.7 ± 2.3 - 30.1 ± 2.2	- 1438 ± 466 - 1391 ± 452	-	-	Time to full enteral feeding, sepsis, NEC, BPD, IVH, mortality, duration of hospital stay
Romagnoli, 2011, Italy [36]	GA ≤ 32 weeks	- 31 - 32	- Lutein (0.5 mg/kg/day) and zeaxanthin (0.02 mg/kg/day) - Placebo	Start: day 7 postpartum, duration: until 40 weeks PMA or until discharge, whichever came first	- 30.0 ± 1.9 - 29.7 ± 1.9	- 1351 ± 438 - 1311 ± 398	ROP	-	Sepsis, NEC, BPD, IVH, mortality, duration of hospital stay
**Enzymes**									
Casper, 2016, (10 European countries) [37]	GA < 32 weeks	- 207 - 208	- RhBSSL added to formula or pasteurized human milk (15 mg/100 mL) - Placebo	Start: enteral nutrition of at least 100 mL/kg/day, duration: 4 weeks	- 28.8 ± 1.7 - 28.8 ± 1.7	- 1179 ± 299 - 1167 ± 294	Growth velocity	Antibodies against rhBSSL	NEC
**Trace elements**									
Aggarwal, 2016, India [38]	GA < 32 weeks and BW < 1500 g	- 55 - 59	- Selenium (10 µg/day) - Placebo	Start: day 1 postpartum, duration: until day 28 of life	- 31.7 ± 0.6 - 31.6 ± 0.6	- 1473 ± 46 - 1455 ± 54	Sepsis	ROP, mortality	Meningitis, IVH, duration of hospital stay
Terrin, 2013, Italy [39]	GA 24–32 weeks or BW 401–1500 g	- 97 - 96	- Zinc (9 mg/day) - Standard care	Start: day 7 postpartum, duration: until 42 weeks PMA or until discharge, whichever came first	- 28 (28–29) ^§^ - 28 (27–29)	- 1114 (1056–1172) ^§^ - 1033 (969–1097)	Sepsis, NEC, BPD, ROP	Mortality, growth velocity	Time to full enteral feeding, IVH, duration of hospital stay
**Compounds of plasma membrane lipids**									
Picaud, 2002, France [40]	Appropriate-for-gestational-age preterm infants	- 10 - 10 - 10	- LD cholesterol (<0.03 g/L) - MD cholesterol (0.15 g/L) - HD cholesterol (0.30 g/L)	Start: at the end of the second week of life, duration: until 40 weeks PMA	- 29.7 ± 1.6 - 29.3 ± 1.8 -29.9 ± 1.6	- 1174 ± 281 - 1209 ± 229 - 1321 ± 281	-	-	Growth velocity
Tanaka, 2013, Japan [41]	BW < 1500 g	- 12 - 12	- Formula with higher dose SM (20% of all phospholipids) - Standard formula (SM 13% of all phospholipids)	Start: <24 h postpartum, duration: not reported	- 29.1 ± 2.1 - 30.1 ± 2.3	- 1116 ± 254 - 1100 ± 353	Neurodevelopment at 6, 12 and 18 months corrected age	-	-
**Creatine**									
Bohnhorst, 2004, Germany [42]	GA < 32, postconceptional age <36 weeks and symptoms of AOP severe enough to require treatment with caffeine	- 19 - 19	- Creatine monohydrate (200 mg/kg/day) - Placebo	Duration: 2 weeks	- 27 (25–30) ^#^ - 27 (25–30)	- 1040 (580–1400) ^#^ - 1015 (440–1525)	-	-	Growth velocity
**Immunoglobulins**									
Lawrence, 2001, Australia [43]	BW ≤ 1500 g	- 768 - 761	- IgG (1200 mg/kg/day) - Placebo	Start: on the day that enteral nutrition was initiated, duration: 28 days	- 28.6 ± 2.6 - 28.4 ± 2.6	-	NEC, mortality	-	-
**Nucleotides**									
Scopesi, 2006, Italy [44]	GA 30–37 weeks	Not reported	- Nucleotide enriched formula - Standard formula	Start: day 2 postpartum, duration: 2 weeks	- 33.7 ± 2.5 - 33.1 ± 1.3	-	-	-	Growth velocity

*** Data are presented as mean ± SD or median (IQR), unless otherwise stated, ^#^ Median (range), ^§^ Mean (95% CI), BW: birth weight, GA: gestational age, VLBW: very low birth weight, PMA: postmenstrual age, LD: low-dose, MD: medium-dose, HD: high-dose, SM: sphingomyelin, RhEPO: recombinant human erytropoetin, RhG-CSF: recombinant human granulocyte colony stimulating factor, RhBSSL: recombinant human bile salt-stimulated lipase, IgG: immunoglobulin G, NEC: necrotizing enterocolitis, ROP: retinopathy of prematurity, BPD: bronchopulmonary dysplasia, IVH: intraventricular hemorrhage.

**Table 2 nutrients-12-02916-t002:** Effects of supplemental enteral bioactive factors in preterm infants.

First Author, Year [Reference] Study Group (Number of Analyzed Infants)	Time to Full Enteral Feeding, Days *	Weight Gain, g/kg/Day *	Head Circumference, cm/Day *	Length, cm/Day *	Culture-Proven Late Onset Sepsis, *n* (%)	Clinical or Suspected Sepsis, *n* (%)	Meningitis, *n* (%)	NEC, *n* (%)	BPD, *n* (%)	ROP, *n* (%)	IVH, *n* (%)	Mortality, *n* (%)	Duration of Hospital Stay, Days *
**Hormones and growth factors**													
El-Ganzoury, 2014 [19]													
RhG-CSF (*n* = 20)	12.6 ± 5.4 ^b^	-	-	-	-	-	-	0 (0) ^g^	-	-	-	2 (10)	44.6 ± 11.9
RhEPO (*n* = 20)	13.4 ± 4.9	-	-	-	-	-	-	0 (0)	-	-	-	2 (10)	43.5 ± 11.1
RhG-CSF + rhEPO (*n* = 20)	12.4 ± 3.1	-	-	-	-	-	-	0 (0)	-	-	-	1 (5)	43.1 ± 9.9
Placebo (*n* = 30)	16.3 ± 5.3	-	-	-	-	-	-	3 (10)	-	-	-	3 (10)	57.9 ± 10.8
*P*-value													
RhG-CSF vs. placebo	0.005	-	-	-	-	-	-	0.165	-	-	-	0.92	<0.001
RhEPO vs. placebo	0.032	-	-	-	-	-	-	0.165	-	-	-	0.92	<0.001
RhG-CSF + rhEPO vs. placebo	0.006	-	-	-	-	-	-	0.165	-	-	-	0.92	<0.001
Omar, 2020 [20]													
RhEPO (*n* = 36)	14 (11–17) ^b^	-	-	-	-	-	-	5 (14) ^e^	-	-	-	15 (42)	-
Placebo (*n* = 36)	15 (11–20)	-	-	-	-	-	-	4 (11)	-	-	-	18 (50)	-
*P*-value	0.708	NS	-	-	-	-	-	1	-	-	-	-	-
Hosseini, 2019 [21]													
AF with rhG-CSF (*n* = 50)	-	-	-	-	12 (24)	-	-	3 (6) ^e^	-	3 (6) ^e^	9 (18) ^e^	2 (4)	-
AF with rhG-CSF + rhEPO (*n* = 50)	-	-	-	-	9 (18)	-	-	3 (6)	-	2 (4)	9 (18)	1 (2)	-
Standard care (*n* = 50)	-	-	-	-	9 (18)	-	-	4 (8)	-	3 (6)	10 (20)	8 (16)	-
*P*-value	-	-	-	-	0.303	-	-	0.763	-	0.741	0.771	0.027	-
Juul, 2003 [22]													
RhEPO (*n* = 15)	-	16.7	-	-	-	-	-	-	-	-	-	-	-
Placebo (*n* = 17)	-	18.4	-	-	-	-	-	-	-	-	-	-	-
*P*-value	-	-	-	-	-	-	-	NS ^†^	-	-	-	-	-
Corpeleijn, 2008 [23]													
Formula with IGF-1 (*n* = 28)	12.6 ± 4.4 ^a^	11	0.095 ± 0.019	-	-	-	-	2 (7) ^e^	-	-	1 (4) ^g^	0 (0)	37 (12–152) ^d,#^
Standard formula (*n* = 32)	12.8 ± 4.2	9	0.093 ± 0.019	-	-	-	-	3 (9)	-	-	2 (6)	3 (9)	30 (9–220)
*P*-value	0.73	0.97	0.93	-	-	-	-	1	-	-	-	0.24	0.94
Shamir, 2013 [24]													
Rh-insulin (*n* = 4)	6.0 (3.5–7.8) ^b^	17.4 (15.5–19.5)	-	-	-	-	-	-	-	-	-	-	-
Placebo (*n* = 4)	13.5 (7.3–16.0)	15.0 (12.7–17.4)	-	-	-	-	-	-	-	-	-	-	-
*P*-value	0.081	0.248	-	-	-	-	-	-	-	-	-	-	-
El-Kabbany, 2020 [25]													
Melatonin (*n* = 20)	-	-	-	-	-	-	-	-	-	-	-	0 (0)	3.0 (2.5–4.5)
Placebo (*n* = 20)	-	-	-	-	-	-	-	-	-	-	-	6 (30)	5.5 (3.5–9.5)
*P*-value	-	-	-	-	-	-	-	-	-	-	-	0.008	0.018
Canpolat, 2006 [26]													
RhG-CSF (*n* = 8)	-	-	-	-	-	-	-	-	-	-	-	-	19.5 ± 3.1
Placebo (*n* = 10)	-	-	-	-	-	-	-	-	-	-	-	-	32.8 ± 3.6
*P*-value	-	-	-	-	-	-	-	-	-	-	-	-	< 0.001
**Vitamins**													
Sun, 2019 [27]													
Vitamin A (*n* = 132)	-	-	-	-	3 (2)	-	-	7 (5) ^g^	25 (19)	2 (2) ^k^	4 (3) ^g^	1 (1)	30.1 ± 6.3
Placebo (*n* = 130)	-	-	-	-	5 (4)	-	-	10 (8)	44 (34)	9 (7)	5 (4)	2 (2)	64.2 ± 7.5
*P*-value	-	-	-	-	0.499	-	-	0.463	0.008	0.034	0.748	0.619	<0.001
Wardle, 2001 [28]													
Vitamin A (*n* = 77)	-	-	-	-	-	-	-	9 (12) ^i^	40 (77)	6 (8) ^m^	4 (5) ^h^	25 (32)	-
Placebo (*n* = 77)	-	-	-	-	-	-	-	7 (9)	37 (77)	6 (8)	7 (9)	29 (38)	-
*P*-value	-	-	-	-	-	-	-	0.58	0.98	0.98	-	-	-
Basu, 2019 [29]													
Vitamin A (*n* = 98)	-	-	-	-	9 (9)	-	-	1 (1) ^e^	2 (2)	1 (1.0) ^n^	4 (4) ^e^	9 (9)	-
Placebo (*n* = 98)	-	-	-	-	18 (18)	-	-	4 (4)	9 (9)	2 (2.0)	7 (7)	16 (16)	-
*P*-value	-	-	-	-	0.021	-	-	0.211	0.05	0.568	0.359	0.141	-
Bell, 2013 [30]													
Vitamin E (*n* = 59)	-	-	-	-	-	-	-	-	-	-	-	-	-
Placebo (*n* = 29)	-	-	-	-	-	-	-	-	-	-	-	-	-
*P*-value	-	-	-	-	NS ^†^	-	-	NS ^†^	-	-	NS ^g^	NS	-
Barekatain, 2018 [31]													
Vitamin E (*n* = 38)	-	-	-	-	3 (8)	14 (37)	-	1 (3) ^†^	-	-	0 (0) ^g^	3 (8)	-
Placebo (*n* = 38)	-	-	-	-	4 (11)	16 (42)	-	3 (8)	-	-	2 (5)	5 (13)	-
*P*-value	-	-	-	-	NS	NS	-	NS	-	-	NS	0.45	-
Pathak, 2003 [32]													
Vitamin E (*n* = 15)	-	-	-	-	5 (33)	-	-	1 (7) ^†^	-	-	-	-	69.7 ± 24.0
Placebo (*n* = 15)	-	-	-	-	5 (33)	-	-	0 (0)	-	-	-	-	69.0 ± 23.2
*P*-value	-	-	-	-	-	-	-	-	-	-	-	-	-
**Carotenoids**													
Dani, 2012 [33]													
Carotenoids (*n* = 58)	-	-	-	-	14 (24)	-	-	3 (5) ^†^	16 (28)	4 (7) ^g^	7 (12) ^†^	3 (5)	-
Placebo (*n* = 56)	-	-	-	-	12 (21)	-	-	3 (5)	15 (27)	7 (13)	6 (11)	4 (7)	-
*P*-value	-	-	-	-	0.903	-	-	0.707	0.909	-	0.946	0.491	-
Manzoni, 2013 [34]													
Carotenoids (*n* = 113)	12.7 ^†^	-	-	-	20 (18)	-	-	2 (2) ^g^	5 (5)	7 (6) ^l^	5 (4) ^g^	4 (4)	-
Placebo (*n* = 116)	14.3	-	-	-	23 (20)	-	-	6 (5)	12 (10)	12 (10)	6 (5)	6 (5)	-
*P*-value	0.25	-	-	-	0.88	-	-	0.15	0.07	0.18	0.9	0.89	-
Costa, 2013 [35]													
Carotenoids (*n* = 38)	-	-	-	-	3 (8)	-	-	1 (3) ^f^	0 (0)	-	1 (3) ^g^	2 (5)	37.0 ± 23.0
Placebo (*n* = 39)	-	-	-	-	5 (13)	-	-	1 (3)	3 (8)	-	2 (5)	1 (3)	41.0 ± 24.0
*P*-value	NS ^b^	-	-	-	1	-	-	1	0.24	-	1	0.616	0.511
Romagnoli, 2011 [36]													
Carotenoids (*n* = 31)	-	-	-	-	3 (10)	-	-	1 (3) ^f^	0 (0)	3 (10) ^g^	1 (3) ^g^	2 (6)	42.7 ± 23.0
Placebo (*n* = 32)	-	-	-	-	4 (13)	-	-	1 (3)	3 (9)	4 (13)	2 (6)	1 (3)	45.3 ± 24.7
*P*-value	-	-	-	-	1	-	-	1	0.238	1	1	0.613	0.616
**Enzymes**													
Casper, 2016 [37]													
RhBSSL (*n* = 206)	-	16.8	-	-	-	-	-	7 (3) ^†^	-	-	-	-	-
Placebo (*n* = 204)	-	16.6	-	-	-	-	-	1 (0.5)	-	-	-	-	-
*P*-value	-	0.493	-	-	-	-	-	-	-	-	-	-	-
**Trace elements**													
Aggarwal, 2016 [38]													
Selenium (*n* = 45)	-	-	-	-	0 (0)	7 (16)	0 (0)	-	-	3 (7) ^e^	-	2 (4)	4.0 ± 2.4 ^d^
Placebo (*n* = 45)	-	-	-	-	6 (13)	16 (36)	3 (7)	-	-	6 (14)	-	3 (7)	5.7 ± 4.4
*P*-value	-	-	-	-	0.03	0.02	0.24	-	-	0.291	-	0.645	0.01
Terrin, 2013 [39]													
Zinc (*n* = 97)	17 (15.3–19.8) ^c,§^	18.2 ± 5.6	-	-	16 (17)	-	-	0 (0) ^g^	10 (10)	0 (0) ^g^	8 (8) ^g^	5 (5)	59 ± 31
Placebo (*n* = 96)	14 (11.6–16.7)	17.0 ± 8.7	-	-	12 (13)	-	-	6 (6)	15 (16)	3 (3)	11 (12)	17 (18)	49 ± 35
*P*-value	0.44	0.478	-	-	0.431	-	-	0.014	0.272	0.121	0.454	0.006	0.181
**Compounds of plasma membrane lipids**													
Picaud, 2002 [40]													
LD cholesterol (*n* = 10)	-	27.1 ± 4.9 ^o^	0.9 ± 0.4	0.7 ± 0.4	-	-	-	-	-	-	-	-	-
MD cholesterol (*n* = 10)	-	32.2 ± 5.6	1.2 ± 0.5	0.8 ± 0.7	-	-	-	-	-	-	-	-	-
HD cholesterol (*n* = 10)	-	30.1 ± 7.7	1.3 ± 0.4	0.7 ± 0.5	-	-	-	-	-	-	-	-	-
*P*-value	-	NS	NS	NS	-	-	-	-	-	-	-	-	-
**Creatine**													
Bohnhorst, 2004 [42]													
Creatine (*n* = 17)	-	17.6 (8.4–31.4) ^#^	-	-	-	-	-	-	-	-	-	-	-
Placebo (*n* = 17)	-	16.4 (4.1–36.4)	-	-	-	-	-	-	-	-	-	-	-
*P*-value	-	NS	-	-	-	-	-	-	-	-	-	-	-
**Immunoglobulins**													
Lawrence, 2001 [43]													
IgG (*n* = 768)	-	-	-	-	-	-	-	50 (7) ^j^	-	-	-	26 (3)	-
Placebo (*n* = 761)	-	-	-	-	-	-	-	47 (6)	-	-	-	22 (3)	-
*P*-value	-	-	-	-	-	-	-	-	-	-	-	-	-
**Nucleotides**													
Scopesi, 2006 [44]													
Nucleotides (*n* = 7)	-	7.9	-	-	-	-	-	-	-	-	-	-	-
Placebo (*n* = 7)	-	3.8	-	-	-	-	-	-	-	-	-	-	-
*P*-value	-	-	-	-	-	-	-	-	-	-	-	-	-

* Data are presented as mean ± SD or median (IQR), unless otherwise stated, ^#^ Median (range), ^§^ Mean (95% CI), ^†^ Definition of the outcome unknown, ^a^ Enteral intake of ≥120 mL/kg/day, ^b^ Enteral intake of ≥150 mL/kg/day, ^c^ Enteral nutrition >110 kcal/kg, ^d^ Duration of stay in the NICU (days), ^e^ Stage ≥ 2, ^f^ Stage >2A, ^g^ Stage ≥ 3, ^h^ Stage 4, ^i^ NEC requiring surgery, ^j^ NEC diagnosed at surgery or by necropsy, or on radiological or clinical grounds, ^k^ Type 1 ROP, ^l^ Threshold ROP, ^m^ Treated ROP, ^n^ ROP requiring laser therapy, ^o^ Growth velocity in g/day. RhEPO: recombinant human erytropoetin, RhG-CSF: recombinant human granulocyte colony stimulating factor, RhBSSL: recombinant human bile salt-stimulated lipase, IgG: immunoglobulin G, LD: low-dose, MD: medium-lose, HD: high-dose, NEC: necrotizing enterocolitis, ROP: retinopathy of prematurity, BPD: bronchopulmonary dysplasia, IVH: intraventricular hemorrhage.

**Table 3 nutrients-12-02916-t003:** Overview of ongoing or recently completed trials investigating the effect of a supplemental enteral bioactive factor in preterm infants.

Intervention Group	Control Group	Intervention Duration	Estimated Number of Participants	Main Inclusion Criteria	Country	Year of Registration	Trial Registry	Trial Registry Identifier
**Hormones and growth factors**								
Insulin (400 µU/mL milk)	Placebo	28 days	33	GA 26–33 weeks	Israel	2010	ClinicalTrials.gov	NCT01093638
Insulin (low-dose group: 400 µU/mL milk, high-dose group: 2000 µU/mL milk)	Placebo	28 days	530	GA 26–32 weeks and BW ≥ 500 g	Belgium, Bulgaria, France, Germany, Hungary, Israel, Italy, The Netherlands, Spain, United Kingdom, United States	2015	ClinicalTrials.gov, EU Clinical Trials Register	NCT02510560, 2014-002624-28
RhEPO (88 IU/kg/day)	Placebo	Until an enteral intake of 150 mL/kg/day or for a maximum of 10 days	72	GA < 32 weeks	Egypt	2018	Pan African Clinical Trial Registry	PACTR201806003426116
Melatonin (3 mg/kg/day)	Placebo	15 days	60	GA < 29 + 6 weeks, able to receive minimal 20 mL/kg/day enteral nutrition within 96 h from birth	Italy	2020	ClinicalTrials.gov	NCT04235673
Melatonin (20 mg)	Standard care	Once	90	GA < 37 weeks, evidence of feeding intolerance	Egypt	2020	ClinicalTrials.gov	NCT04304807
**Vitamins**								
Vitamin A (5000 IU/kg/day)	Placebo	28 days	914	GA < 32 weeks, BW < 1000 g, < 72 h of age, oxygen supplementation or respiratory support in the first 72 h of life	Germany, Austria	2014	EU Clinical Trials Register, German Clinical Trials Register	2013-001998-24, DRKS00006541
Vitamin A (30,000 IU/kg/day) [45]	Standard care	6 weeks	209	GA ≤ 32 weeks and BW ≤ 1250 g, FiO2 > 21% within the first 24 h of life	Turkey	2014	Conference abstract in Archives of Disease in Childhood	P0-0731
Vitamin A (5000 IU/day) [46]	Placebo	Start: within 24 h of initiation of enteral feeding, duration: until 34 weeks’ PMA	188	GA < 28 weeks and <72 h of life	Australia	2016	Australian New Zealand Clinical Trials Registry	ACTRN12616000408482
Vitamin A (2000 IU/day) and vitamin D (700 IU/day)	Standard care	28 days	976	GA < 32 weeks, <96 h of age	China	2018	ClinicalTrials.gov	NCT03779776
Vitamin E (12.5 IU every 12 h)	Placebo	Start: 72 h of life, duration: until day 28 postpartum	90	BW < 1500 g, diagnosed with RDS, mechanical ventilation or CPAP	Mexico	2017	ClinicalTrials.gov	NCT03274596
**Carotenoids**								
Lutein (0.5 mg/kg/day) and zeaxantin (0.05 mg/kg/day)	Placebo	Start: <36 h of life, duration: until day 30 postpartum	50	GA ≤ 32 weeks and/or BW ≤ 1500 g	Italy	2017	ClinicalTrials.gov	NCT03340103
**Trace elements**								
Zinc (1.4 mg/kg/day)	Standard care	10 days	180	Preterm infants with sepsis	Egypt	2020	Thai Clinical Trials Registry	TCTR20200624002
Zinc (10 mg/day)	Standard care	Until discharge	120	GA 28–37 weeks with sepsis	India	2017	Clinical Trials Registry India	CTRI/2017/08/009544
Zinc acetate (2 mg/kg/day)	Standard care	Through 36 + 6/7 weeks PMA	126	GA 23–30 weeks, BW 501–1000 g, 14–28 days of life, 14-day BPD risk score ≥50% for death or moderate-severe BPD	United States	2018	ClinicalTrials.gov	NCT03532555
Zinc (10 mg/day)	Placebo	Start: day 3 postpartum, duration: 40 weeks PMA or discharge, whichever comes first	364	GA 28–32 weeks	Indonesia	2019	ClinicalTrials.gov	NCT04050488

GA: gestational age, BW: birth weight, PMA: postmenstrual age, BPD: bronchopulmonary dysplasia.

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
