# Peer review of "Enteral Bioactive Factor Supplementation in Preterm Infants: A Systematic Review"

_nutrients, 2020, doi:10.3390/nu12102916_

Round 1

Reviewer 1 Report

This systematic review describes the importance of some bioactive compounds in mother's milk that can favour preterm infants in a very comprehensive way. It is well written and I do not have any remarks to make.

Many bioactive molecules in breast milk are often overlooked, and some of their properties are ill-known to many specialists. This review summarizes the most important existing literature, and equally points out shortcomings therein, with claims strengthend by meta-analysis. Moreover, the paper gives nice suggestions on what should be tried out next. Hence, all necessary keypoints of a good review are addressed, at least in my viewpoint.

Reviewer 2 Report

The authors report a systematic review in the effect of bioactive factors administered enterally in preterm mortality and morbidity. No significant effect is reported based in the small sample size of the studies and the heterogeneity of the interventions. Methodological the systematic review is well conducted.

The manuscript is inspiring for neonatologists and nutritionists. Databases for currently ongoing studies have been also searched. Vitamin supplementation is the most potent topic of the review. Tables are informative and well designed. Discussion is nicely written. More discussion in sample size, absorption and biodisponibility of lipid compounds and ongoing studies is welcome.
